# Direct Mercury Detection in Landfill Leachate Using a Novel AuNP-Biopolymer Carbon Screen-Printed Electrode Sensor

**DOI:** 10.3390/mi12060649

**Published:** 2021-06-01

**Authors:** Jae-Hoon Hwang, David Fox, Jordan Stanberry, Vasileios Anagnostopoulos, Lei Zhai, Woo Hyoung Lee

**Affiliations:** 1Department of Civil, Environmental, and Construction Engineering, University of Central Florida, Orlando, FL 32816, USA; 2NanoScience Technology Center and Department of Chemistry, University of Central Florida, Orlando, FL 32816, USA; foxd@knights.ucf.edu (D.F.); lzhai@ucf.edu (L.Z.); 3Department of Chemistry, University of Central Florida, Orlando, FL 32816, USA; Stanberry.jordan@knights.ucf.edu (J.S.); Vasileios.Anagnos@ucf.edu (V.A.)

**Keywords:** Au nanoparticle, biopolymer, co-electrode position, landfill leachate, square wave anodic stripping voltammetry (SWASV)

## Abstract

A novel Au nanoparticle (AuNP)-biopolymer coated carbon screen-printed electrode (SPE) sensor was developed through the co-electrodeposition of Au and chitosan for mercury (Hg) ion detection. This new sensor showed successful Hg^2+^ detection in landfill leachate using square wave anodic stripping voltammetry (SWASV) with an optimized condition: a deposition potential of −0.6 V, deposition time of 200 s, amplitude of 25 mV, frequency of 60 Hz, and square wave step voltage of 4 mV. A noticeable peak was observed at +0.58 V associated with the stripping current of the Hg ion. The sensor exhibited a good sensitivity of ~0.09 μA/μg (~0.02 μA/nM) and a linear response over the concentration range of 10 to 100 ppb (50–500 nM). The limit of detection (LOD) was 1.69 ppb, which is significantly lower than the safety limit defined by the United States Environmental Protection Agency (USEPA). The sensor had an excellent selective response to Hg^2+^ in landfill leachate against other interfering cations (e.g., Zn^2+^, Pb^2+^, Cd^2+^, and Cu^2+^). Fifteen successive measurements with a stable peak current and a lower relative standard deviation (RSD = 5.1%) were recorded continuously using the AuNP-biopolymer-coated carbon SPE sensor, which showed excellent stability, sensitivity and reproducibility and consistent performance in detecting the Hg^2+^ ion. It also exhibited a good reliability and performance in measuring heavy metals in landfill leachate.

## 1. Introduction

Mercury (Hg) pollution caused by industrial activity has been attracting global attention for decades [1,2,3]. Mercury occurs in many forms in aqueous solution depending on oxidation and reduction conditions [4]. The main forms of Hg exposure in the general population include methylmercury (MeHg) from seafood, inorganic mercury (I–Hg) from food, and mercury vapor (Hg^0^) from dental amalgam fillings [5]. Most Hg occurs in organic and inorganic forms of divalent mercury and Hg^0^, as a form of Hg dissolved in an aqueous phase [6]. Once Hg has reached surface waters or soils, microorganisms convert it to MeHg, a substance that can be absorbed quickly by most organisms including marine life. MeHg, upon consumption, can cause negative health effects (i.e., nerve, kidney and intestinal damage; stomach disruption; reproductive failure; and DNA alteration) [7]. Inorganic mercury, Hg^0^ and Hg^2+^, is released into the environment from a variety of anthropogenic and natural sources. In particular, Hg^2+^ ions are one of the largest hazardous Hg pollutants in aquatic ecosystems [8]. Furthermore, Hg^2+^ is not readily biodegradable and is prone to bioaccumulation and biomagnification across trophic levels. Exposure to Hg^2+^ has been linked to several diseases; such as Minamata [9], acrodynia [10], cardiac [11] and neurological disorders, [12] and several developmental illnesses.

Surface water and groundwater are often polluted by untreated sewage water, industrial waste, gasoline, medical waste, electroplated steel and electronic parts. Since Hg accumulates in the ecosystem during tropospheric cycling, it is essential to regulate the presence of Hg^2+^ in drinking water and environmental systems. The United States Environmental Protection Agency (USEPA) has mandated an upper limit of 2 ppb (10 nM) of Hg^2+^ in drinking water [3], so monitoring of very low concentrations of Hg in the early stages of pollution is required to assess hidden risks. Various analytical methods such as atomic absorption spectrophotometry (AAS) [13], atomic fluorescence spectrometry (AFS) [14], inductively coupled plasma mass spectrometry (ICP-MS) [15], and X-ray fluorescence spectrometry [16] have been developed for detecting Hg. These are generally combined with cold vapor generation and amalgamation techniques to separate and pre-concentrate Hg to achieve a high sensitivity [17,18]. However, the complicated pretreatment requires a variety of different reagents and numerous preparation steps. Electrochemical techniques are very effective in detecting low concentrations of mercury because metal ions can be rapidly pre-concentrated on the electrode surface using methods such as anodic stripping voltammetry (ASV) [19]. In addition, modifying electrodes can offer ease of operation and low-cost and portable instrumentation [20]. The sensitivity and selectivity of Hg electrochemical sensors have been greatly improved by nanostructured electrodes, which provide a large surface area and intriguing properties [21,22,23].

Gold nanostructures have attracted great research interest in the voltammetric detection of mercury because of their high affinity [24], good catalytic ability [25], high adsorption capacities [26], and excellent mass transport [27,28]. Among the many methods to prepare AuNPs, the reduction of chloroauric acid (HAuCl_4_) is the most popular [29], but recently, the use of polymers to synthesize AuNPs has been reported. For example, AuNPs were prepared in the presence of chitosan [21,22] and its derivative carboxymethylated chitosan [24]. Chitosan is recognized as a low-cost promising supplement to AuNPs and other metal film electrodes due to its relatively high mechanical strength, good adhesion to traditional electrochemical surfaces, high water permeability, biocompatibility and [30,31] ability to form stable chelates with transition metal ions [32]. However, the application of chitosan-coated AuNPs has been limited to specific applications due to their tendency to agglomerate and precipitate; therefore coupling with a suitable supporting substrate may increase their viability in preparation of metal composite sensors [33].

In this study, we demonstrated direct Hg^2+^ detection in real landfill leachate samples using a AuNP-biopolymer-coated carbon SPE sensor. AuNP and the biopolymer (chitosan) were coated onto a carbon SPE sensor by electrodeposition and gave enhanced sensing performance for Hg^2+^ detection through an increase in both electrochemical sensitivity and stability. Using the new sensor, systematic batch experiments were performed to determine the optimal deposition time amplitude and frequency for detecting Hg^2+^ ions using square wave anodic stripping voltammetry (SWASV). The sensor performance for th eon-site monitoring of Hg^2+^ including calibration curves, potential interference, repeatability, recovery, and limit of detection (LOD) was then fully evaluated in a landfill leachate matrix. This study was the first to investigate direct electrochemical Hg^2+^ detection in a real wastewater sample.

## 2. Materials and Methods

### 2.1. AuNP-Chitosan Electrode Sensor Fabrication

AuNP-chitosan composite film was electrochemically deposited on a carbon screen-printed electrode (SPE) (RRPE1001C, Pine Research Instrumentation, Durham, NC, USA) as a working electrode. The nanocomposite solution contained chloroauric acid (HAuCl_4_) and chitosan, based on the optimization ratio of chitosan to sensor performance used in a previous study [34]. The schematic illustration of the electrodeposition of the composite material is presented in Appendix A. The AuNP and biopolymer nanocomposite was mixed at a 1:1 ratio (12 mg chitosan in 10 mL DI solution and 0.01 M HAuCl_4_ solution), which was stirred for 24 h at 60 °C to achieve complete mixing. Then, 3 µL of the solution was dropped onto the surface of the carbon SPE sensor and completely deposited by electrodeposition at 6.4 mA/cm^2^ DC for 1 h. The electrodeposition was repeated three times to achieve sufficient deposition of AuNPs. The fabricated sensor was cleaned with DI water, dried at room temperature, and stored under ambient conditions before use.

### 2.2. Structure Characterization of AuNP-Biopolymer Composite

The surface structures of the fabricated SPE sensors were obtained using an Ultra 55 scanning electron microscope (SEM) (ZEISS, Oberkochen, Germany). Elemental mapping of the surface was achieved through energy-dispersive X-ray spectroscopy (EDS) using a Noran System 7 EDS with a silicon drift X-ray detector (Thermo Scientific™, Waltham, MA, USA). To characterize the surface chemical and oxidation states, X-ray photoelectron spectroscopy (XPS) was performed by a Thermo Scientific ESCALAB Xi+ X-ray Photoelectron Spectrometer Microprobe (Thermo Scientific™, Waltham, MA, USA) with a twin-crystal, micro-focusing monochromator and an Al anode. The XPS analysis chamber during measurement was held at a pressure below 1.0 × 10^−7^ torr. Electrochemical impedance spectroscopy (EIS) spectra were measured using a PalmSens 3 EIS potentiostat (PalmSens Compact Electrochemical Interfaces, Houten, The Netherlands) under the three-electrode system in 10 mM potassium ferricyanide (K_3_Fe(CN)_6_) solution at the frequency range 5 Hz to 50 kHz.

### 2.3. Chemicals and Sample Preparation

Analytical grade sodium acetate was obtained from Sigma-Aldrich (St. Louis, MO, USA) and used without further purification. To characterize the developed sensor for Hg detection, a 0.1 M acetate buffer solution (AcB) at pH 3.0 was used. We used a stock Hg solution (1 mg/mL in nitric acid, 7487-94-7, Sigma-Aldrich, St. Louis, MO, USA) ranging from 2 to 20 ppb in the electrolyte (i.e., acetate buffer). For the sensor evaluation in real wastewater, landfill leachate samples were obtained from a local landfill (Orange county landfill, Orlando, FL, USA). The characterization of the landfill leachate samples is shown in Appendix A. Test water samples were prepared by spiking precalculated amounts of Hg^2+^ ranging from 10 to 100 ppb into the landfill leachate and the sensor performance of mercury detection was evaluated for various mercury concentrations. To adjust pH to 3, 0.1 M HCl was added to the landfill leachate sample. The samples were kept in room conditions of 23 °C with a relative humidity of 45%. The prepared Hg^2+^ test solutions were validated using a single quadrupole ICP-MS (iCAP-RQ, Thermo Scientific, Waltham, MA, USA).

### 2.4. Sensor Characterization and Electrochemical Mercury Detection

A 20 mL electrochemical cell (Compact Voltammetry Cell-Starter Kit, Pine Research Instrumentation, Durham, NC, USA) with an effective volume of 10 mL was used for Hg^2+^ detection, and each series was measured by a three-electrode system consisting of an AuNP-chitosan working electrode, carbon counter electrode (RRPE1001C, Pine Research Instrumentation, Durham, NC, USA), and Ag/AgCl reference electrode (MI-401, Microelectrodes, Inc., Bedford, NH, USA). We used a PalmSens 3 EIS (PalmSens Compact Electrochemical Interfaces, Houten, The Netherlands) as a potentiostat for all tests (Figure 1). Mercury detection using the developed AuNP-biopolymer SPE sensor was characterized using SWASV and CV. First, CV was performed at a scan rate of 50 mV/s in 0.1 M AcB (pH 3.0) with Hg^2+^ (50 ppb) to determine the potential window of the AuNP-biopolymer SPE sensor for Hg^2+^ detection and the potential where the Hg^2+^ stripping peak appears. For the Hg^2+^ measurement using SWASV, Hg^2+^ was electrochemically deposited on the working electrode at −0.8 V at pH 3.0 of 0.1 M AcB for 100 s, and the reduced Hg^2+^ was then stripped according to predetermined parameters (4 mV step potential, 25 mV amplitude, and 20 Hz frequency). For sequential measurements, the working electrode was cleaned at +0.8 V for 60 s to remove remnants of Hg^2+^ ion before the next measurement. All tests were conducted in triplicate with a value of mean ± standard deviation (SD). The LOD was calculated based on three times the signal-to-noise (S/N ratio of 3) using the Equation (1): C_L_ = kS_B_/b [35], where C_L_ signifies the detection limit, S_B_ represents the standard deviation of blank signals, k is a parameter with a value of 3, and b is the value of the calibration curve slope.

## 3. Results and Discussion

### 3.1. Characterization and Modification of AuNP-Biopolymer Nanostructure

The surface analysis by SEM showed that the electrode surface was covered with gold nano-urchins (AuNPs) as well as some small AuNPs, as seen in Figure 2. The AuNPs were on the order of 1 μm in diameter, with many spines on the surface of each urchin-like particle. The AuNPs shown on the electrode surface were in the tens of nanometers in diameter. Elemental analysis with energy-dispersive X-ray spectroscopy (EDS) mapping confirmed that the location of the Au signal was concentrated in these structures on the electrode surface. The EDS spectra (Appendix A) showed prominent characteristic X-ray peaks from C K_α_ and Au M emissions. In addition to the increased surface area, the Au urchin-like structures were expected to have improved performance through several mechanisms. For example, a greater portion of this exposed surface is likely composed of high-index facets. These have been shown to increase catalytic performance [36,37] and with Miller indices (*hkl*) (where at least one value is greater than 1) have a greater number of low-coordinated atoms on the surface [38]. These facets have higher surface energy, and as a result, the deposition takes place preferentially on these facets during particle growth. Co-electrodeposition of Au and chitosan successfully generated these urchin-like structures in addition to the free AuNPs on the surface of the SPE sensor.

The surface chemical state was determined by XPS, as shown in Figure 3. Deconvolution of the C 1s and N 1s core-level peaks confirmed the surface adhesion of the biopolymer. The C–C peak at 284.7 eV was the dominant peak in the sample, with a signal contribution from both the chitosan polymer and the exposed regions of the carbon electrode. Deconvolution analysis shows a C–N peak at 286.4 eV and a C–O peak at 287.2 eV, which are consistent with the structure of chitosan. The peak at 289.1 eV, consistent with C=O moiety, suggests some acetylated portions of the chitosan chain or residual carbon monoxide gas. Due to the low signal/noise ratio in the N 1s region, the deconvolution of the N 1s band is less certain; deconvolution may indicate an amide moiety peak at 400.0 eV, with −NH_2_ and −NH_3_ peaks at 399.3 eV and 400.7 eV, respectively, confirming a deacetylated chitosan biopolymer on the surface. Au 4f peaks were located at 84.5 eV (4f_7/2_) and 88.3 eV (4f_5/2_), with full width at half maxima (FWHM) of 0.80 eV and 0.76, respectively. The peak and FWHM are consistent with metallic Au [39].

### 3.2. AuNP-Biopolymer-Coated Carbon SPE Sensor Characterization

The electron transfer properties of the developed AuNP-biopolymer-coated carbon SPE sensor were studied by electrochemical impedance spectra (EIS) (Figure 4a). A carbon SPE sensor coated only with chitosan, one coated only with AuNPs, a bare carbon SPE sensor, and a bare gold SPE sensor were also prepared and compared for reference purposes. The charge transfer resistance for the chitosan-coated carbon SPE sensor (398 Ω) was higher compared to that of the bare carbon SPE sensor (228 Ω) and the bare gold SPE sensor (70 Ω), indicating that the chitosan only-coated carbon SPE sensor had hindered charge transfer from the redox probe of [Fe(CN)_6_]^3−/4−^ to the surface of the fabricated electrode. In contrast, the Nyquist plots of the AuNP-coated SPE sensors (i.e., AuNP-coated carbon SPE sensor and AuNP-biopolymer-coated carbon SPE sensor) displayed an almost straight line, showing a very fast charge transport process due to direct electron transfer. This implied that the presence of AuNP nanocomposites enabled the enhancement of electron transfer kinetics suitable for achieving a superior sensor response. The value of the charge transfer resistance also depended on the dielectric properties at the electrode/electrolyte interface [40].

To investigate sensor characteristics for the oxidation and reduction of Hg^2+^ by varying potentials, CV was performed in 0.1 M AcB with a 50 ppb Hg^2+^ concentration (Figure 4b). The corresponding redox peaks current was gradually increased in the following order: chitosan-coated carbon SPE sensor, AuNP coated carbon SPE sensor, and AuNP-chitosan-coated carbon SPE sensor. The highest redox peak current was obtained with the AuNP-chitosan-coated carbon SPE sensor, which was attributed to the improvement in electron transport due to the metallic Au nanoparticles in chitosan. Lower redox peaks were observed for the bare electrode series (carbon and gold) due to the hindrance of electron transport because of the absence of the Au NP/chitosan composite on the working electrode. The observed findings were consistent with the results obtained from the EIS measurements. These results gave immediate evidence that the modification of the sensing interface successfully detected Hg^2+^.

### 3.3. Optimization of SWASV Parameters for Hg^2+^ Detection

To achieve high performance in Hg^2+^ detection, the electrochemical parameters for SWASV (deposition potential, deposition time, amplitude, and frequency) were optimized at a fixed Hg^2+^ concentration of 20 ppb in 0.1 M AcB at pH 3.0. First, the effect of the deposition potential on Hg^2+^-stripping peak currents was investigated between −1.0 and −0.2 V with a deposition time of 100 s, the amplitude of 25 mV and a frequency of 20 Hz [34] (Figure 5a). The current of 4.0 ± 0.1 μA at −0.6 V of deposition potential was 1.2–1.7 fold higher than other deposition potentials between −0.2 V and −1.0 V. The resulting SWASV peaks showed the maximum stripping peak current at −0.6 V, indicating the greatest reduction of Hg^2+^. At deposition potentials higher than −0.6 V, the current obtained from stripping Hg^2+^ decreased as the deposition potential increased, probably due to the inefficient deposition of Hg^2+^. On the other hand, at reduction potentials below −0.6 V, the reduction of Hg^2+^ was less efficient because the reaction began to compete with H_2_ generation, which typically occurs below this potential [41,42]. As such, it could be seen that the stripping peak current decreased at a deposition potential lower than −0.6 V based on the CV tests. A similar trend of deposition potential was also obtained by Rahman et al. (2019) [43]. Thus, a deposition potential of −0.6 V was chosen as an optimum potential for Hg^2+^.

Deposition time is known to affect the amount of Hg^2+^ ions [43] deposited on the AuNP-chitosan electrode, thus influencing the LOD and the overall time needed for the SWASV technique. The peak current of anodic Hg^2+^ stripping increased from 10.4 ± 0.6 to 39.7 ± 1.3 μA with a prolonged deposition time at the optimized deposition potential of −0.6 V (Figure 5b). The peak current for Hg^2+^ initially increased with deposition time, then slightly increased after 200 s probably due to Hg^2+^ saturation on the electrode surface. Although the sensitivity can be improved with a longer deposition time, surface saturation at high metal ion concentrations can also reduce the upper detection limit [44]. Correspondingly, the deposition time of 200 s was chosen as optimal for the fast detection of Hg^2+^ with high selectivity.

The effects of amplitude and frequency on th current response to Hg^2+^ were also investigated (Figure 5c,d). Optimal conditions for amplitude were observed at 25 mV with the highest peaks at 4.3 ± 0.04 μA. Increasing the amplitude range from 0.05 to 0.1 V resulted in higher ambient noise with a larger SD (Appendix A) due to vibrations [45]. For frequency optimization, the peak heights increased from 20 to 100 Hz, while a higher frequency showed larger ambient noise after 80 Hz (Appendix A) because of the frequency properties of vibration [45]. Therefore, 60 Hz was chosen as the optimal frequency for Hg^2+^ detection using the AuNP-chitosan SPE sensor. Overall, −0.6 V of deposition potential, 200 s of deposition time, 25 mV amplitude, and 60 Hz of frequency were selected as optimal SWASV analytical parameters for further evaluation of the sensor performance on Hg^2+^ detection.

### 3.4. Sensitivity Analysis and Lifetime

Figure 6a shows the SWASV response of AuNP-biopolymer-coated carbon SPE sensor at various Hg^2+^ concentrations between 0–20 ppb in 0.1M AcB (pH 3.0) under optimized parameters. The well-defined sharp anodic stripping peaks for detecting Hg^2+^ were located at +0.58 V, and the peak currents were linearly increased with Hg^2+^ concentrations. The applied optimal parameter of the AuNP-biopolymer-coated carbon SPE sensor exhibited improved sensor sensitivity (17× more sensitive than an unoptimized one) with a higher R^2^ value (R^2^ = 0.9914), indicating enhanced sensitivity and stability for Hg^2+^ detection (Appendix A). The correlation was I_p_ = 0.477x + 0.0009, where I_p_ is the stripping peak current (µA) and x is Hg^2+^ concentration (ppb) (Figure 6b). The sensitivity of the sensor was ~0.1 µA/nM (0.477 µA/µg) according to the slope of the linear curve. The LOD of the AuNP-biopolymer-coated carbon SPE sensor toward Hg^2+^ was estimated to be 0.9 ppb according to Equation (1), which is well below the USEPA-defined limit for drinking water (2 ppb) [3]. The relative standard deviation (RSD) value of 2.27% was evaluated by replicate measurements (n = 20) of 20 ppb Hg^2+^ solution (Figure 6c). The linear range was also 25× larger than that of other SPE sensors (bare gold and bare carbon) (Appendix A). A comparison between the developed AuNP-biopolymer-coated carbon SPE sensor and other electrochemical Hg-detecting sensors, based on a gold electrode or AuNPs previously reported in the literature, is listed in Table 1. The AuNP-biopolymer-coated carbon SPE sensor had a lower Hg^2+^ detection range compared to those of previous studies. This is attributable to the AuNP surface, where free functional groups of amino acids (i.e., chitosan) exhibits a strong affinity for Hg^2+^ between Au and Hg [46]. The sensor also showed relatively lower RSD and detection limits.

### 3.5. Selectivity Analysis: Interference of Other Heavy Metals in Hg^2+^ Detection

Metal ions commonly found in water containing Hg^2+^ are known to compromise the accuracy of electrochemical measurements [48]. Thus, the influence of competing metal ions was investigated in two-fold concentrations over the analyte (i.e., 40 ppb) interfering species (Zn^2+^, Cd^2+^, Pb^2+^, and Cu^2+^) in 20 ppb Hg^2+^ in 0.1 M AcB (pH 3.0). Figure 7 shows a similar range of sensor responses in the presence of these heavy-metal ions. Most of these (Zn^2+^, Cd^2+^, and Pb^2+^) did not interfere with the detection of Hg^2+^. A slight decrease in the current value (~8.7%) was shown in the presence of Cu^2+^, a major interfering species because of its close potential to Hg^2+^ measurement [52]. To further validate this, we performed an experiment by mixing ions with a fixed concentration of Hg^2+^ at 20 ppb. No decrease in peak current or interfering voltammetric peaks was observed in the solution. The developed AuNP-biopolymer-coated carbon SPE sensor was then applied to the Hg^2+^ detection in landfill leachate as an example of a practical analytical application.

### 3.6. Applications for Landfill Leachate Environment

To evaluate sensor behavior in a real wastewater matrix, landfill leachate samples were collected on site (Orange County Landfill, Orlando, FL, USA) without pretreatment except for filtration through 0.45 μm Polyethersulfone (PES) filters. The pH of the landfill leachate water was ~7.7 and included high heavy-metal and ionic concentration (Appendix A). The pH was adjusted to 3.0 using 0.1 M HCl to have predominant species of Hg^2+^ in the range of pH 1–3 [53]. The original landfill leachate sample was directly used for the baseline curve and electrolyte due to a high NaCl concentration (13,800 µS/cm). Application of the AuNP-biopolymer-coated carbon SPE sensor in this experiment resulted in well-defined peaks at +0.6V for a range of Hg^2+^ concentrations, 0–100 ppb, for the landfill leachate, under which, the anodic peak potential shifted towards the positive due to the positive electrocatalytic activity of the Hg^2+^ ions with the sensor. This phenomenon is common as adsorption products tend to shift the peak potential positively [54]. Hg^2+^ sensitivity and the LOD of AuNP-biopolymer-coated carbon SPE sensor showed 0.089 μA/ppb (R^2^ = 0.993) and 1.69 ppb, respectively (Figure 8a,b). The reproducibility experiments conducted using landfill leachate spiked with Hg^2+^ (50 ppb) achieved 15 stable, successive measurements (Figure 8c) with an RSD of 5.1%. The concentrations of Hg^2+^ in the spiked landfill leachate sample were validated using ICP-MS and compared with the results of our method (Table 2). The recoveries were in the range of 98–108%, indicating an acceptable performance for practical applications.

## 4. Conclusions

We demonstrated direct Hg^2+^ detection in landfill leachate using a newly developed AuNP-biopolymer carbon SPE sensor. The combination of the good conductivity of AuNP-chitosan and its strong adhesion to Hg^2+^ gave the sensor high sensitivity and selectivity for Hg^2+^ determination with a detection limit of 0.9 ppb in 0.1 M AcB. The sensor was successfully applied to determine Hg^2+^ in real landfill leachate samples directly with recovery ranging from 98 to 108%. The LOD and RSD for 15 consecutive measurements of the fabricated sensor in the landfill leachate samples were 1.69 ppb and 5.1%, respectively. The minimal interference in the presence of other heavy metal ions was observed in the detection of Hg^2+^ ions using the sensor. Overall, the developed AuNP-biopolymer carbon SPE sensor is expected to demonstrate reliable Hg^2+^ sensing in wastewater including landfill leachate.

## Figures and Tables

**Figure 1 micromachines-12-00649-f001:**
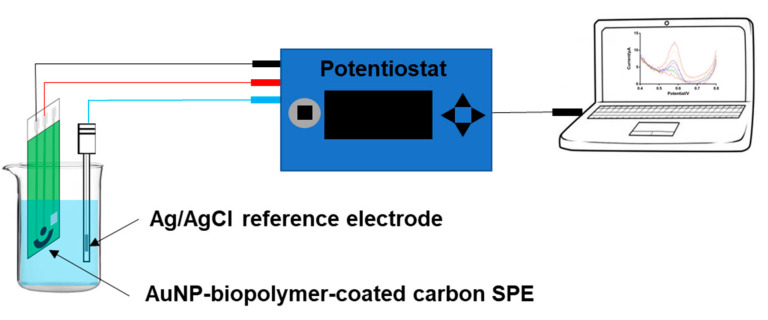
Measurement setup schematic for testing an AuNP-biopolymer-coated carbon SPE.

**Figure 2 micromachines-12-00649-f002:**
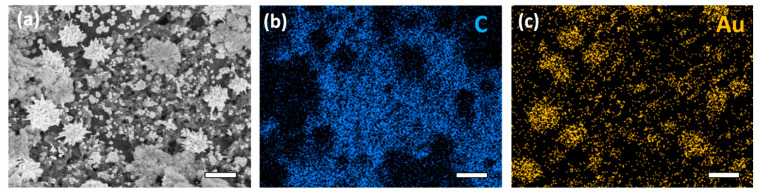
SEM and EDS mapping of AuNP-biopolymer composite film electrodeposited on a carbon electrode. The electron map obtained with the secondary detector is shown (**a**), with the corresponding carbon intensity map (**b**), and gold intensity map (**c**). All scale bars are 1 μm.

**Figure 3 micromachines-12-00649-f003:**
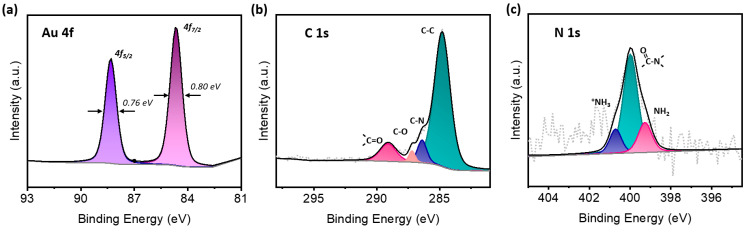
XPS of electrodeposited AuNP-biopolymer nanocomposite. (**a**) Au 4f spectra, (**b**) C 1s spectra, and (**c**) N 1s spectra.

**Figure 4 micromachines-12-00649-f004:**
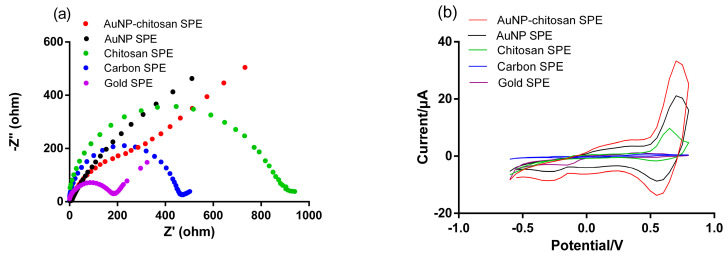
(**a**) Nyquist diagrams and (**b**) CV responses of a bare carbon SPE sensor, a bare gold SPE sensor, a biopolymer modified carbon SPE sensor, and an AuNP-biopolymer nanocomposite SPE sensor.

**Figure 5 micromachines-12-00649-f005:**
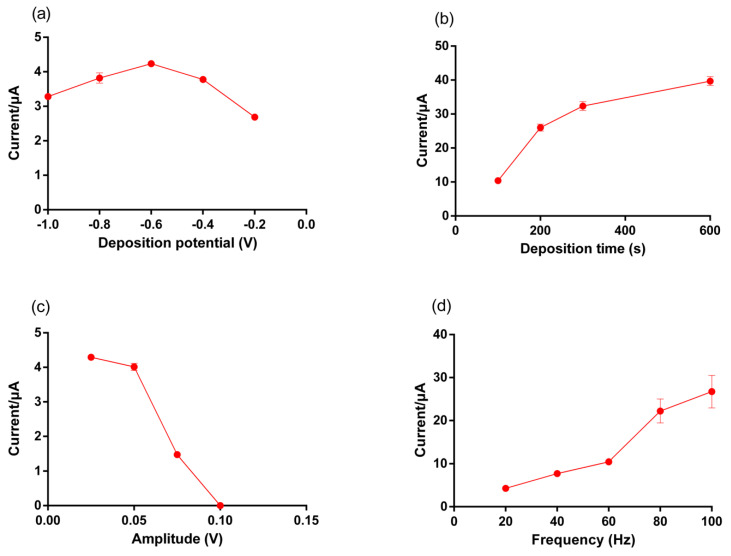
Optimization of (**a**) deposition potential, (**b**) deposition time, (**c**) amplitude and (**d**) frequency on the stripping peak currents of Hg^2+^ using an AuNP-biopolymer-coated SPE sensor. Hg^2+^ concentration was 10 ppb (0.1M AcB at pH 3.0). Individual data points represent the standard deviation (±SD) of duplicate experiments.

**Figure 6 micromachines-12-00649-f006:**
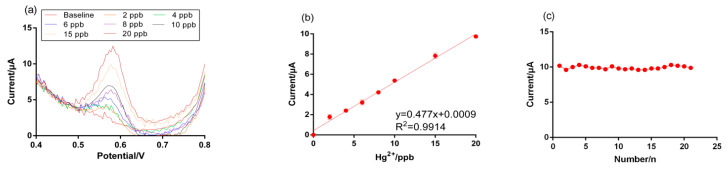
(**a**) SWASV replies for Hg^2+^ determination, (**b**) plot of the stripping peak current vs Hg^2+^ concentration, and (**c**) reproducibility. Deposition time is 200 s with a −0.6 V deposition potential, 0.004 V potential step, 0.025 V amplitude, and 60 Hz frequency. Hg^2+^ concentration was 20 ppb (0.1M acetate buffer at pH 3.0). The error bars represent the standard deviation for the mean of the three replicate tests.

**Figure 7 micromachines-12-00649-f007:**
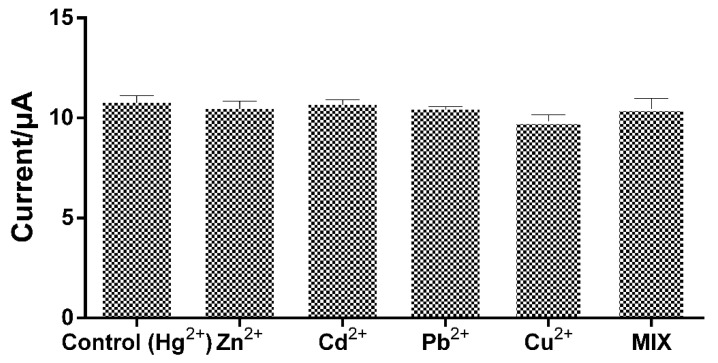
Sensor response changes at a fixed Hg^2+^ concentration (20 ppb) in the presence of other metal ions (40 ppb) in 0.1 M AcB (pH 3.0). Mix indicates the solution containing Zn^2+^, Cd^2+^, Pb^2+^, and Cu^2+^ (40 ppb) along with Hg^2+^ (20 ppb). The error bars represent the standard deviation for the mean of the three replicate tests.

**Figure 8 micromachines-12-00649-f008:**
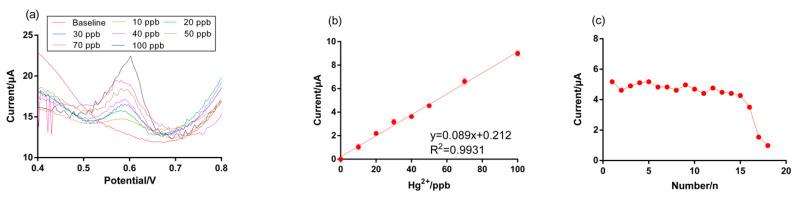
Hg^2+^ detection using a AuNP-biopolymer-coated carbon SPE sensor in a real landfill leachate environment (**a**) SWASV, (**b**) calibration curves, and (**c**) reproducibility (Hg^2+^ concentration is 50 ppb in landfill leachate (pH 3.0)). The error bars represent the standard deviation for the mean of the three replicate tests.

**Table 1 micromachines-12-00649-t001:** Comparison of different types of sensor performance for electrochemical Hg (II) detection.

Electrode	Method	Linear Range	LOD (nM)	RSD (%)	Reproducibility (n)	Sample Condition (Buffer Solution)	Reference
AuNPs/CFME ^(1)^	DPASV ^a^	1–250 µM	0.5	3.4	−	0.1 M HCl/pH 1	[47]
np-AuNPs/ITO ^(2)^	DPASV	0.5–50 nM	0.15	2.3	7	0.1 M HCl/pH 1	[24]
Au-DMAET-(SWCNT-PABS) ^(3)^	SWASV ^b^	20–250 µM	63.4	2.7	10	0.1 M HCl/pH 3	[48]
SWCNT-PhSH/Au ^(4)^	SWASV	5–90 nM	3.0	3.8	7	0.1 M HCl/pH 1	[49]
SPGE ^(5)^	SWASV	5–30 µM	5.5	4.3	40	0.1 M HCl/pH 2	[50]
AuNPs-GC ^(6)^	CV ^c^	0.64–4 µM	0.42	−	−	0.01 M HCl/pH 2	[29]
Cys-AuNPs-CILE ^(7)^	SWASV	10–20,000 nM	2.3	2.6	5	0.1 M phosphate/pH 7	[51]
AuNP-biopolymer-coated carbon SPE sensor	SWASV	10–100 nM	4.5	2.3	20	0.1 M acetate/pH 3	This study

^a^ DPASV (differential pulse anodic stripping voltammetry); ^b^ SWASV (square wave anodic stripping voltammetry); ^c^ CV (cyclic voltammetry); ^(1)^ Gold Nanoparticles (AuNPs)/three-dimensional fibril-like carbon-fiber mat electrode (CFME); ^(2)^ Nanoporous gold nanoparticles (np-AuNPs)/Indium tin oxide (ITO); ^(3)^ Au-dimethyl amino ethanethiol (DMAET)-Single-walled carbon nanotube-poly (m-amino benzene sulfonic acid) (SWCNT-PABS); ^(4)^ Single-walled carbon nanotube (SWCNTs) with thiophenol/Gold (Au); ^(5)^ Screen-printed gold electrodes (SPGE); ^(6)^ Gold nanoparticles–modified glassy carbon (AuNPs-GC); ^(7)^ l-cysteine (Cys)-Au nanoparticle-Carbon ionic liquid electrode (CILE).

**Table 2 micromachines-12-00649-t002:** Comparison of Hg^2+^ detection methods between AuNP-biopolymer-coated carbon SPE sensor and ICP-MS (n = 3).

Sample	Hg^2+^ Concentration			Recovery (%)
	Add (ppb)	Detection (ppb)		
		AuNP-Biopolymer-Coated Carbon SPE Sensor	ICP-MS	
Landfill leachate 1	15	14.7 ± 1.8	15.2 ± 0.9	98
Landfill leachate 2	20	21.6 ± 3.4	20.6 ± 1.3	107.8
Landfill leachate 3	30	31.5 ± 2.1	30.8 ± 1.4	105

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
