# Peer review of "Direct Mercury Detection in Landfill Leachate Using a Novel AuNP-Biopolymer Carbon Screen-Printed Electrode Sensor"

_micromachines, 2021, doi:10.3390/mi12060649_

Round 1

Reviewer 1 Report

The article is well prepared and of practical interest. I think it can be published after some revision. I would recommend that the authors add the sensor and measurement schematics in section 2, which would be helpful to the readers. Also it would be useful to see the frequency response of the sensor. What is the reproducibility of the sensor responses, how many samples were used?

Reviewer 2 Report

The SWASV responces of the proposed electrode are not acceptable. They sufffer from noise (Fig 5). Besides the concentration range of Hg is similar with other more simpler gold SPE. I am not in the position to understand the true advantages of this presented  "noisy" sensor. What about the inteference of Ag? For the above reasons, I am not in the position to suggest this manuscript for publication in Micromachines. 

Reviewer 3 Report

Authors have demonstrated the direct Hg2+ detection in landfill leachate using an AuNP-biopolymer carbon SPE sensor. The manuscript is relatively well written and analyzed. There is room to improve the figure qualities such as missing legends and labels, and borrowed equations should be cited where applicable. Authors have used the terms sensitivity and detection limit many times in the text. Have they defined them? If not authors should define (quantitively) these parameters relevant to the field and compare them with the limit of detection defined by the United States Environmental Protection Agency. 

Reviewer 4 Report

Hwang and his co-workers, in this work, presented the construction of a mercury sensor, based on an Au nanoparticle (AuNP)-biopolymer coated carbon screen-printed electrode, for direct detection of mercury in landfill leachate. The surface structures of the fabricated SPE sensors were characterized by SEM, EDS and XPS methods. Sensor properties were tested by determining the limit of detection, linearity range, sensitivity, selectivity, stability, potential interference of other heavy metal ions, and repeatability. Performance of the sensor was fully evaluated in landfill leachate matrix. The research is interesting and carefully done. The manuscript is well-written. I suggest authors consider comments below:

--- Please include in line 70 after the “The sensitivity and selectivity of Hg electrochemical sensors have been greatly improved by nanostructured electrodes that provide large surface area and intriguing properties [21,22]” The following relevant review as additional reference: Recent Advances in Sensing Application ……… Adv. Funct. Mater. 2017, 27, 1702891; DOI: 10.1002/adfm.201702891.

--- Figure 2. c) The signal/noise for the N 1s region is very small, and not surprisingly, the fitting of a curve to the experimental spectrum is not too good; consequently, the deconvolution of the N 1s band is uncertain. Authors should indicate this in lines 186 and 187 by stating for example: ‘Due to the low signal/noise in the N 1s region, the deconvolution of the N 1s band is less certain; deconvolution may indicate an amide moiety peak at 400.0 eV, with -NH2 and -NH3 peaks at 399.3 eV and 400.7 eV respectively, confirming the deacetylated chitosan biopolymer on the surface.’

--- Line 217: replace ‘with an order’ with ‘in the order’.

--- Lines 287-289 and Table 1: Please modify the following statement “other electrochemical Hg detecting sensors previously reported in the literature” to “other electrochemical Hg detecting sensors based on gold electrode or AuNPs, previously reported in the literature, is’. There are many more Hg sensors in the literature than you cite; you find reference to assembled graphene based Hg sensors, for example, in the review cited above.

Round 2

Reviewer 2 Report

The revised version of the manuscript is very improved. I have the following suggestion before the acceptance of this manuscript for publication. The SWASV voltammograms of Fig 6 a should be presented under optimal parameters (deposition potential,  time, amplitude,  frequency) with reduced noise.